# Protein S100-A7 Derived from Digested Dentin Is a Critical Molecule for Dentin Pulp Regeneration

**DOI:** 10.3390/cells8091002

**Published:** 2019-08-29

**Authors:** Shungo Komichi, Yusuke Takahashi, Motoki Okamoto, Manahil Ali, Masakatsu Watanabe, Hailing Huang, Takeo Nakai, Paul Cooper, Mikako Hayashi

**Affiliations:** 1Department of Restorative Dentistry and Endodontology, Osaka University Graduate School of Dentistry, 1-8 Yamadaoka, Suita-shi, Osaka 565 0871, Japan; 2Osaka Research Institute of Industrial Science and Technology, 1-6-50 Morinomiya, Jyoutou-ku, Oosaka-shi, Osaka 536 8853, Japan; 3Oral Biology, School of Dentistry, University of Birmingham, Edgbaston, Birmingham B15 2TT, UK

**Keywords:** wound healing, pulp capping materials, dentinogenesis, proteomics

## Abstract

Dentin consists of inorganic hard tissue and organic dentin matrix components (DMCs). Various kinds of bioactive molecules are included in DMCs and some of them can be released after digestion by endogenous matrix metalloproteinases (MMPs) in the caries region. Digested DMCs induced by MMP20 have been reported to promote pulpal wound healing processes, but the released critical molecules responsible for this phenomenon are unclear. Here, we identified protein S100-A7 as a critical molecule for pulpal healing in digested DMCs by comprehensive proteomic approaches and following pulp capping experiments in rat molars. In addition, immunohistochemical results indicated the specific distribution of S100-A7 and receptor for advanced glycation end-products (RAGE) as receptor for S100-A7 in the early stage of the pulpal healing process, and following accumulation of CD146-positive stem cells in wounded pulp. Our findings indicate that protein S100-A7 released from dentin by MMP20 might play a key role in dentin pulp regeneration.

## 1. Introduction

The dentin-pulp complex has been reported to possess a natural regenerative potency [1]. The progression of dental caries and/or stimulation of cavity preparation can lead to exposure of pulp tissue with local necrosis or alterations of the underlying odontoblasts. After such degeneration of the dentin-pulp complex, repair or regenerative processes can occur with newly formed hard tissue called “tertiary dentin” [2,3]. Tertiary dentin formation is regarded as a result of a successful wound healing process within the pulp including cell proliferation, odontogenic differentiation, and mineralized tissue formation [1,2]. Direct pulp capping is the only treatment option to promote this regenerative process, but the long-term success rate is reported to be approximately 60%–70% [4,5]. To improve the success rate, many investigations have used growth factors, transcription factors, or other molecules as pulp capping agents [6,7,8]. However, an evidence-based pulp capping material is not available at present because the wound healing mechanism of pulp tissue is still unclear.

In various injured tissues such as skin, heart, lung, and bone, matrix metalloproteinases (MMPs) play important roles during wound healing processes digesting the extracellular matrix (ECM) of the tissues [9,10,11]. MMPs digest the ECM enzymatically and release ECM fragments including growth factors and cytokines to facilitate the tissue repair process [12,13,14]. In the case of the dentin-pulp complex, dental caries or trauma can induce activation of MMPs in dentin and pulp [15,16,17]. Subsequently, these MMPs digest organic dentin matrix components (DMCs), which are regarded as a bioactive ECM for pulp cells, and trigger their release or activation. For example, transforming growth factor (TGF)-β, which has various functions and is present in both dentin and bone [18,19], is released from the ECM and activated by MMPs through proteolytic degradation of the latent TGF-β complex [20,21]. The same phenomenon has been reported for dentin matrix protein-1 [22,23] and dentin sialophosphoprotein [24,25,26]. Enzymatically digested DMCs have been reported to have functional effects on pulp cells [27], and we have reported that DMCs digested by MMPs may stimulate wound healing of the dentin-pulp complex in a similar manner as that in the abovementioned organs [15,28].

In our previous study, DMCs digested by MMP20 promoted wound healing processes of dentin-pulp complexes both in vitro and in vivo [28]. There may be numerous bioactive molecules including growth factors, cytokines, and fragments of other matrix proteins released from DMCs digested by MMP20. Some of them play critical roles in tertiary dentinogenesis by promotion of biological repair processes. However, the details of these digested products and the crucial molecules exerting bioactive effects on pulp cells are unclear. Therefore, we considered that identification of the critical bioactive molecules among digested products was necessary to elucidate the mechanism of the wound healing processes of pulp and the following tertiary dentinogenesis. In addition, elucidation of the wound healing mechanism might be exploited to develop a new pulp capping agent that would lead to a higher and stable success rate of pulp capping treatments and novel therapeutic approaches for dental tissue regeneration based on the biological wound healing process. In the present study, we attempted to identify the bioactive molecule in DMCs after digestion by MMP20 using proteomic approaches, which promoted wound healing processes of the dentin-pulp complex.

## 2. Materials and Methods

### 2.1. Preparation of DMCs

Human dentin specimens were collected from extracted non-carious teeth at the Oral Surgery Department of Birmingham Dental Hospital (UK). All experimental procedures were performed in accordance with the relevant guidelines and regulations of the Ethics Committee of the University of Birmingham School of Dentistry Tooth Bank (Approval no. 90/H0405/33) as described in a previous report [27]. Informed consent was obtained from all subjects.

Preparation of DMCs has been described previously [28,29]. In brief, powdered human dentin was demineralized in a 10% EDTA solution with proteinase inhibitors, 10 mM n-ethylmaleimide (Sigma-Aldrich, St. Louis, MO, USA), and 5 mM phenyl-methyl-sulphonyl fluoride (Sigma-Aldrich), for 14 days at 4 °C. Then, the eluted supernatants containing organic components were collected and lyophilized after dialysis against repeated changes of distilled water for a further 10 days. The acquired materials were regarded as DMCs. One milligram of DMCs was dissolved in 500 μL phosphate buffered saline (PBS) and incubated with 2 µg MMP20 (Funakoshi, Tokyo, Japan) for 1 day at 37 °C, which were called digested DMCs (d-DMCs). DMCs incubated without MMP20 were prepared as the control and regarded as untreated DMCs (u-DMCs). The incubation time and concentration of MMP20 were determined by previous experiments [28].

### 2.2. Reverse-Phase High Performance Liquid Chromatography (RP-HPLC) Analysis and Fraction Collection

Fifty microliters of u-DMCs or d-DMCs were analyzed by RP-HPLC (LC-10ADVP system with an SPD-M10A UV/Vis detector; Shimadzu, Kyoto, Japan) for 100 min of elution with a 0.05% formic acid aqueous solution as the mobile phase buffer at a flow rate of 0.1 mL/min using a COSMOSIL PBr Packed Column (4.6 mm I.D. × 250 mm; Nacalai Tesque, Kyoto, Japan) monitored at 254 nm. Fractions showing a specific peak or peaks were collected manually by the fraction collector, and fractions with no peaks were collected every 10 min in order not to miss the proteins, which could not be detected as a peak by the UV/Vis detector at this wavelength. The collecting period was 3–10 min without peaks (Figure 1A).

The total volume of each collected fraction was adjusted to 2 mL by adding the 0.05% formic acid aqueous solution.

### 2.3. Animal Studies Using Fractions of u-DMCs or d-DMCs

#### 2.3.1. Pulp Capping Experiment

All experimental procedures were approved and performed in accordance with the relevant guidelines of the Ethical Guidelines Committee for Animal Care of Osaka University Graduate School of Dentistry (Approval no. 23-005-1).

The acquired fractions diluted 50-fold with PBS were prepared for pulp capping. Eight-week-old male Wistar rats (Clea Japan, Inc., Tokyo, Japan) were used in the pulp capping experiment. After intraperitoneal anesthesia induced by pentobarbital (Kyoritsuseiyaku, Tokyo, Japan) and carprofen (Pfizer, New York, NY, USA), the upper first molar was isolated by the rubber dam technique and disinfected by alcohol as part of the aseptic procedure. Pulp exposure of approximately 0.2 mm in diameter was established on the occlusal surface of the tooth using a round steel bur (0.8 mm diameter; Dentsply Maillefer, Ballaigues, Switzerland) as described previously [15]. After cleaning the cavity with a 2.5% sodium hypochlorite solution and saline, a gelatin sponge (Spongel^®^; Astellas Pharma Inc., Tokyo, Japan) containing 20 μL of each diluted sample (d-DMC fractions #1–14) was gently applied as a pulp capping material, and then the cavity was sealed with glass ionomer cement (Fuji IX; GC, Tokyo, Japan). The 0.05% formic acid aqueous solution, which was used as the mobile phase buffer in HPLC analysis, was diluted 50-fold with PBS and used as the control. After 4 weeks, the rats were perfused with physiological saline followed by 4% paraformaldehyde (Nacalai Tesque), and their teeth were evaluated histologically to screen for bioactive fractions (*n* = 3 for each group).

Fractions #12 and #13 were mixed and adjusted to 2 mL with the 0.05% formic acid aqueous solution. The mixture of fractions #12 and #13 from u-DMCs or d-DMCs diluted 50-fold with PBS was used for pulp capping experiments in the same manner as described above. The formic acid aqueous solution was used as the control. After 4 weeks, the volume of newly formed hard tissue was quantified using a µ-CT scanner (R_mCT2; Rigaku, Tokyo, Japan) as described in a previous study [28], and histological evaluation was performed.

#### 2.3.2. Histological Evaluation

After additional fixation in 4% paraformaldehyde for 12 h, the specimens were decalcified in a 10% formic acid/citric acid solution and embedded in paraffin, and then 5 μm-thick sections were prepared. These sections were stained with hematoxylin-eosin (H-E) and evaluated.

#### 2.3.3. Tertiary Dentin Volume Quantification by µ-CT Analysis

To quantify the volume of newly formed tertiary dentin, the obtained specimens were scanned by the µ-CT scanner at 90 kV and 160 μA as described previously [28].

For µ-CT analysis, the *X*-axis was set in parallel with the floor of the pulp chamber, the *Y*-axis was set in parallel with the line connecting buccal and lingual roots, and the *Z*-axis was set in parallel with the line connecting medial and distal roots. The radio-opaque area just beneath the pulp exposure site was regarded as newly formed hard tissue in the XY plane.

We referred to the CT value of normal dentin to select the newly formed hard tissue area. This area was measured in a binarized image, and the volume of newly formed hard tissue was calculated by integrating all measured areas. Image analysis software (TRI 3D-BON; Ratoc System Engineering, Tokyo, Japan) was used for quantification with reference to previous reports [30,31].

### 2.4. Cell Studies

#### 2.4.1. Cell Isolation and Culture

Pulp tissue was collected from incisors of 6-week-old male Wistar rats, minced, and then trypsinized using trypsin-EDTA (Sigma-Aldrich). Isolated cells were cultured in α-minimum essential medium (α-MEM; Gibco, Thermo Fisher Scientific, Waltham, MA, USA) supplemented with 20% fetal bovine serum (FBS; Sigma Aldrich) and 10 μg/mL penicillin-streptomycin (Sigma-Aldrich) at 37 °C in a humidified incubator with 5% CO_2_ using cell culture dishes (Becton Dickinson and Company, Franklin Lakes, NJ, USA). Culture medium was changed every 3 days until the cells reached 80% confluence. The rat primary pulp cells (RPPCs) were used in the following in vitro experiments.

#### 2.4.2. WST-1 Assay

Cell metabolism was measured by a WST-1 assay. RPPCs were seeded at 5000 cells/well in a 96-well plate in culture medium (α-MEM supplemented with 1% FBS and 10 μg/mL penicillin-streptomycin) containing fractions #12 and #13 from u-DMCs or d-DMCs, or the 0.05% formic acid aqueous solution as the control (the volume percentage of each specimen was adjusted to 2% (*v*/*v*)).

After 5 days of culture, WST-1 regent (Premix WST-1; Takara Bio, Inc., Shiga, Japan) was added to the culture medium, followed by incubation for 2 h at 37 °C in a humidified incubator with 5% CO_2_. Optical density was measured at 450 nm in a spectrophotometer with a microplate reader (ARVO MX; Perkin-Elmer, Waltham, MA, USA) according to the manufacturer’s instructions (*n* = 6).

#### 2.4.3. Alkaline Phosphatase (ALP) and Alizarin Red Staining

ALP activity was measured by ALP staining, and mineralized tissue was detected by alizarin red staining methods.

RPPCs were seeded at 50,000 cells/well in a 24-well plate in calcifying medium (α-MEM supplemented with 10% FBS, 10 μg/mL penicillin-streptomycin, 50 μg/mL ascorbic acid, and 10 mM sodium beta-glycerophosphate) containing fractions #12 and #13 from u-DMCs or d-DMCs, or the 0.05% formic acid aqueous solution as the control (the volume percentage of each sample was adjusted to 2%(*v*/*v*)).

ALP activity was evaluated after 12 days of culture. The cells were fixed in a 10% formaldehyde neutral buffer solution (KISHIDA CHEMICAL, Osaka, Japan), and ALP staining was performed using an Alkaline Phosphatase Staining Kit (Cosmo Bio, Tokyo, Japan) according to the manufacturer’s instructions. Images were obtained under an inverted microscope (ECLIPSE TS100; Nikon, Tokyo, Japan). The captured images were binarized, and the stained areas were calculated by ImageJ software (NIH, Bethesda, MD, USA) [32], (*n* = 6).

Mineralized tissue was evaluated after 3 weeks of culture. After fixation in the 10% formaldehyde neutral buffer solution, alizarin red staining was performed using an alizarin red staining kit (PG Research, Tokyo, Japan). Alizarin red dye was extracted with 5% formic acid, and the absorbance at 405 nm was measured by the ARVO MX microplate reader (*n* = 6).

### 2.5. Proteomic Analysis

We performed liquid chromatography-tandem mass spectrometry (LC-MS/MS) and then protein identification of fractions #12 and #13 from u-DMCs and d-DMCs. In addition to the above analyses, protein abundance was calculated by constructing an extracted ion chromatogram.

Tryptic peptides were acquired with 0.1% formic acid. Digested samples were loaded on an LC-20AD nanoHPLC (Shimadzu, Japan) by the autosampler onto a C18 trap column (200 µm I.D. × 2.0 cm; BGI, Shenzhen, China), and the peptides were eluted onto a resolving analytical C18 column (75 µm I.D. × 10 cm; BGI). Data acquisition was performed with a TripleTOF 5600 System (AB SCIEX, Concord, ON, Canada) fitted with a Nanospray III source (AB SCIEX) and pulled quartz tip as the emitter (New Objectives, Woburn, MA, USA). The acquired spectrum was analyzed by peptide mass fingerprinting using MaxQuant version 1.5.3.8 (Max Planck Institute of Biochemistry, Martinsried, Germany) and UniProt assuming trypsin as the digestion enzyme. Initial peptide mass tolerance and fragment ion mass deviations were set to 20 ppm and 0.5 Da, respectively. Variable modification (methionine oxidation and N-terminal acetylation) and fixed modification (cysteine carbamidomethylation) were set, and one missed cleavage was allowed for the search. The minimum peptide length was set to seven amino acids, and the false discovery rate (FDR) for peptide and protein identification was controlled at a low level (FDR < 0.01).

### 2.6. Direct Pulp Capping Using Identified Proteins

Human recombinant protein S100-A7 (PROSPEC, Ness-Ziona, Israel) or protein S100-A8 (PROSPEC) was applied to exposed pulp tissue as a pulp capping material by the same procedure described in “Animal studies using fractions of u-DMCs or d-DMCs”. In brief, each recombinant protein was diluted in PBS and adjusted to 1 μg/mL. A gelatin sponge containing 20 μL of the prepared protein samples was gently applied as a pulp capping material. As a control, we used a gelatin sponge containing 20 μL PBS. After 4 weeks, the form of newly formed hard tissue was evaluated using the µ-CT scanner, and the same histological evaluation was performed as described above (*n* = 3 for each group).

### 2.7. In Vivo Damaged Pulp Model and Immunohistochemical Staining

Eight-week-old male Wistar rats were used to establish the damaged pulp model. After general anesthesia, a cavity was prepared on the mesial surface of the upper first molar without pulp exposure using the round steel bur. The cavity depth was controlled up to half of the original dentin and checked by the µ-CT scanner. After 1, 3, and 7 days, the rats were perfused with physiological saline followed by 4% paraformaldehyde. After additional fixation in 4% paraformaldehyde for 12 h, the specimens were decalcified in Kalkitox (Fujifilm, Tokyo, Japan), embedded in paraffin, and cut into 4 μm-thick sections. These sections were evaluated by H-E staining as described above and immunohistochemical staining.

For immunohistochemical staining, the sections were deparaffinized and rehydrated, and then antigen retrieval was performed using citrate buffer (10 mM citric acid, pH 6.0). An ABC Kit (Vector Laboratories, Burlingame, CA, USA) was used according to the manufacturer’s instructions with a rabbit anti-protein S100-A7 polyclonal antibody diluted at 1:20 (Abcam, Cambridge, UK), rabbit anti-receptor for advanced glycation end-products (RAGE) polyclonal antibody diluted at 1:100 (Abcam), and rabbit anti-CD146 monoclonal antibody diluted at 1:400 (Abcam). For final visualization of the sections, a DAB staining kit (Vector Laboratories) was used. The immunostained sections were counterstained with hematoxylin.

### 2.8. Statistical Analysis

All data are expressed as the mean ± standard error of the mean (s.e.m.) of triplicate determinations. Data were analyzed for statistical significance by one-way analysis of variance with the Tukey–Kramer test using Statview software (SAS Institute Inc., Cary, NC, USA) (α = 5%).

## 3. Results

### 3.1. HPLC Analysis and Fraction Collection

Peak analysis visualized by RP-HPLC was performed to separate components in aggregated d-DMC constructs consisting of various kinds of molecules and to isolate the bioactive fractions. Sample-specific peaks were detected in u-DMCs and d-DMCs (Figure 1A). Different shaped peaks were found at around 90 min of retention time as indicated by arrows in Figure 1A. In total, 14 fractions were collected from each sample (Figure 1A).

### 3.2. Determination of Bioactive Fractions In Vivo

To determine which fractions from d-DMCs included bioactive molecules, a direct pulp capping experiment in rat molars was carried out using the 14 fractions collected from d-DMCs by RP-HPLC.

Representative histological images at 28 days after pulp capping are shown in Figure 1B. Pulp-exposed areas were completely covered by newly formed tertiary dentin (TD in Figure 1B) using fractions #12 or #13 from d-DMCs. These peaks showed a sample-specific shape and difference in peak analysis. In contrast, no or little hard tissue was observed when using the diluted mobile buffer as a control for pulp capping (Figure 1B). The other fractions, #1–#11 and #14, from d-DMCs also resulted in no or little hard tissue formation (see Appendix A).

To confirm that the dentinogenesis using fractions #12 or #13 from d-DMCs (Figure 1B) was induced by d-DMC-specific digested products, we compared the effects of fractions #12 and #13 from d-DMCs with those of the same fractions from u-DMCs by direct pulp capping experiments. In µ-CT analysis, newly formed tertiary dentin was shown as pink stereographic images (Figure 2A). The pulp exposure site was perfectly covered by tertiary dentin only in the specimen using the fractions of d-DMCs. Quantification of the tertiary dentin volume showed that the fractions of d-DMCs significantly promoted a larger volume of hard tissue formation compared with the same fractions of u-DMCs and the control (*p* < 0.05, Figure 2B). Histological evaluation of d-DMC-treated specimens showed a tubular-like structure in the predentin area and column-shaped odontoblast-like cells (Figure 2D).

### 3.3. Effects of the Fractions on RPPCs In Vitro

Effects of both fractions #12 and #13 on cell metabolism were evaluated by a WST-1 assay. After five days of culture, the fractions from d-DMCs and u-DMCs significantly promoted the metabolism of RPPCs compared with the control (*p* < 0.05, Figure 3A). There was no significant difference between the fractions of d-DMCs and u-DMCs.

Fractions #12 and #13 from d-DMCs induced significantly higher ALP activity at 12 days of culture compared with the control and the same fractions of u-DMCs (*p* < 0.05, Figure 3B). The fractions of d-DMCs also significantly promoted mineralized tissue formation at 21 days of culture compared with the control and the same fractions of u-DMCs (*p* < 0.05, Figure 3C).

### 3.4. Identification of the Bioactive Molecules

Results of protein identification by LC-MS/MS of fractioned samples are shown in Figure 4A. MaxQuant software, which is a set of algorithms with high peptide identification rates and high-accuracy protein quantification, was used in this study [33].

Thirty-two proteins were detected in fractions #12 and #13 of u-DMCs, and 51 proteins were detected in the same fractions of d-DMCs. Twenty-nine proteins were detected as common proteins in the fractions of u-DMCs and d-DMCs. Among these 29 commonly detected proteins, nine proteins showed a high protein ratio (>1) (Figure 4A). These results indicated that the nine proteins were detected with higher abundance in d-DMCs compared with u-DMCs. The 22 proteins unique to d-DMCs and nine common proteins were considered as candidate bioactive molecules that promoted tertiary dentinogenesis (Appendix A), because the fractions of d-DMCs were clarified to be bioactive in the above in vivo and in vitro experiments. Based on gene ontology (GO) analysis and previous reports, four proteins were identified as candidate critical molecules for pulp repair (Figure 4B). Junction plakoglobin [34] and protein S100-A7 [35,36] were selected from the commonly detected proteins. Protein S100-A8 [36] and prolactin-inducible protein [37] were detected in the unique fractions of d-DMCs. All four proteins had sufficient peptide scores. Posterior error probability of the identification value is essentially considered as a p-value in MaxQuant, where smaller is more significant (Table 1). The values of the four proteins were less than 0.05 in this study.

Among the four proteins, we focused on proteins S100-A7 and S100-A8 in the following experiments, which are regarded as components of dentin [38,39] and bioactive proteins.

### 3.5. Effects of Proteins S100-A7 and S100-A8 on Tertiary Dentinogenesis

Recombinant protein S100-A7 or S100-A8 was applied to exposed pulp as a pulp capping material with a gelatin sponge.

Four weeks later, µ-CT analysis and histological staining revealed that both proteins induced tertiary dentin formation (Figure 5). Tertiary dentin completely covering exposed pulp was observed after applying S100-A7, whereas a tunnel defect was seen in tertiary dentin formed using S100-A8 (white arrows in Figure 5A and black arrows in Figure 5B). In addition to a high potential for tertiary dentin formation, protein S100-A7 exerted an antimicrobial effect that is advantageous for pulp regeneration in the oral environment. Therefore, protein S100-A7 released from dentin might be more implicated in the pulpal healing process, and we focused on this protein in the following experiments.

### 3.6. Expression of Receptor for Advanced Glycation end Products (RAGE) in Damaged Pulp

To confirm the above speculation, an in vivo damaged pulp model was studied. The pulpal wound healing process was observed histologically (Figure 6B), and the expression patterns of the protein S100-A7 RAGE, which is a known receptor for S100-A7, and CD146 were investigated immunohistochemically in the model (Figure 6C–E). The cavity depth was measured by using µ-CT analysis (Figure 6A).

As shown in Figure 6B, odontoblasts had become detached from the predentin beneath the cavity on day 1 (black arrow heads in Figure 6B). Although a newly differentiated odontoblast-like cell layer had begun to reattach to the dentin on day 3 (white arrow heads in Figure 6B), the width of the predentin-like low calcified dentin area was wider (white arrows in Figure 6B) compared with predentin at the apical side (black arrows in Figure 6B). On day 7, newly formed tertiary dentin was observed beneath the cavity (# on Figure 6B). An organized odontoblast-like cell layer was observed along the pulp-tertiary dentin border.

As shown in Figure 6C, the protein S100-A7 was detected mainly at the border between predentin and the detached odontoblasts on day 1 after cavity preparation. A section of the odontoblasts expressed S100-A7 beneath the cavity. On day 3, protein S100-A7-positive cells were clearly detected in the nuclei of pulp cells at the affected area. On day 7, the staining has increased but the nuclei have become difficult to be distinguished. Figure 6D showed distribution of RAGE. RAGE-positive cells were increased on day 3 and their number was decreased on day 7. The distribution of CD146 expression is shown in Figure 6E. In the pulp horn area of the non-affected side, which was considered as an internal control, CD146-positive cells were arranged in the sub-odontoblastic area at even intervals. On day 1, CD146 was observed along the expanded blood vessels. On day 3, an increasing number of CD146-positive cells as well as vasculogenesis were observed. On day 7, accumulation of CD146-positive cells was observed adjacent to the tertiary dentin. Notably, a proportion of the CD146-positive area (black arrows in Figure 6E) coincided with some of the RAGE-positive area (black arrows in Figure 6D).

## 4. Discussion

When caries progression leads to necrosis or alterations of the underlying odontoblasts, undifferentiated pulp cells (dental pulp stem cells; DPSCs) respond to the alterations in the ECM and induce regeneration of lost odontoblasts and dentin via wound healing processes. Whereas digestion of the dentin matrix and subsequent release of bioactive molecules have been reported to facilitate the wound healing process of pulp tissue [15,27], the details of the released molecules were unknown. In the present study, we identified the critical molecules released from DMCs digested by MMP20, which may promote wound healing processes in the dentin-pulp complex.

DMC preparations have an autolytic activity for proteins because of certain enzymes in DMCs [40], suggesting a synergistic action of MMP20 and other proteolytic enzymes in DMCs. In this study, u-DMCs were incubated for 24 h in the same manner as d-DMCs without addition of MMP20 as a control.

u-DMCs and d-DMCs were separated into 14 fractions by RP-HPLC to acquire the fractions containing bioactive molecules. As a result, all peaks were observed within a 100 min retention time at every wavelength by the ultraviolet detector. This result is in agreement with a previous study reporting the effects of dentin-derived protein fractions analyzed by RP-HPLC on human periodontal ligament cells [41,42]. Different shaped peaks at around 90 min of retention time indicated that different molecules were contained in fractions #12 and #13 of u-DMCs and d-DMCs (Figure 1A).

Based on the above results of HPLC analysis, the results of the direct pulp capping experiment using the specific fractions were reasonable (Figure 1B), and fractions #12 and #13 from d-DMCs were found to include bioactive molecules. However, these molecules might also exist in u-DMCs. To confirm that they were d-DMC-specific bioactive molecules released by digestion of DMCs, the effect of the combination of fractions #12 and #13 of both samples on tertiary dentinogenesis was evaluated in vivo. Tertiary dentin formation was evaluated by µ-CT analysis that has two merits for tertiary dentin evaluation. First, more sufficient objectivity is gained than conventional sectional histological methods in terms of detecting dentin defects. Tertiary dentin can be evaluated three-dimensionally, consecutively, and objectively [28,43]. The other merit is that quantification of the tertiary dentin volume is possible. As a result, fractions #12 and #13 from d-DMCs promoted a greater amount of tertiary dentin with no defects compared with the fractions from u-DMCs (Figure 2A,B). In addition to µ-CT analysis, histological evaluation is necessary to evaluate the repair response of pulp tissue and confirm the results of µ-CT evaluation. In the histological evaluation of this study, an odontoblast-like cell layer was observed beneath the newly formed tertiary dentin that had a tubular-like structure in the predentin area (Figure 2C,D). These results indicated different components in the fractions of u-DMCs and d-DMCs despite collection at the same retention time. Therefore, d-DMC-specific bioactive molecules were released by dentin digestion and eluted into fractions #12 and #13 of d-DMCs via RP-HPLC, which promoted tertiary dentin formation. All fractions used for pulp capping experiments contained 0.001% formic acid because a 0.05% formic acid aqueous solution was used as the mobile phase buffer in HPLC analysis. Formic acid at around this concentration has no effect on cellular functions [41,42]. A gelatin sponge was used as a carrier in this study, and this material itself is inert and has no effect on wound healing of hard tissue [44]. Therefore, the molecules contained in the specific fractions of d-DMCs probably promoted the wound healing process of pulp.

The in vivo results were verified by in vitro experiments (Figure 3). The fractions of d-DMCs or u-DMCs significantly promoted cell metabolism compared with the control. ALP activity and mineralized tissue formation were significantly promoted only by the fractions of d-DMCs compared with the same fractions of u-DMCs or the control. Unlike the present results, a previous study has reported that incubating pulp cells with u-DMCs significantly promotes ALP activity and mineralization in cells [45]. In addition to the different concentrations used in our experiments, condensation or exclusion of some molecules by HPLC could be the reasons for this inconsistency. Our results indicated that the specific fractions of d-DMCs promoted cell proliferation, odontoblast differentiation, and dentinogenesis.

LC-MS/MS analysis was performed to identify candidate critical molecules for pulpal repair in the specific fractions of d-DMCs. In the Venn diagrams (Figure 4A), 22 unique proteins from d-DMCs and nine common proteins with a high protein ratio, which indicated high abundance proteins in d-DMCs, were regarded as candidate molecules for these bioactive effects. Because the fractions of d-DMCs had greater effects than those of u-DMCs in vivo, more critical molecules may be included in d-DMCs than u-DMCs. Four proteins out of these 31 proteins were identified as candidate critical molecules for dentin pulp regeneration because they are annotated with “molecular function” as the GO categorization in UniProt (http://www.uniprot.org/) and reported to promote cellular functions that facilitate tissue regenerative events. Among the identified four proteins, proteins S100-A7 and S100-A8 have already been detected in dentin by previous comprehensive proteomic studies of teeth [38,39].

Protein S100 family members are low molecular weight proteins with two calcium-binding sites and are involved in the regulation of a wide range of cellular processes [46,47]. In addition to their intercellular roles, they are also regarded as members of damage-associated molecular patterns (DAMPs) and activate a nuclear factor-κB (NF-κB)-mediated proinflammatory and regenerative response in the ECM [35,47]. In the case of teeth, some S100 family members are detected in carious-affected pulp more abundantly than in normal pulp [48]. This finding indicates that protein S100 family members may perform important roles in pulpal wound healing. Interestingly, proteins S100-A7 and S100-A8 were clarified to promote tertiary dentin formation in the present study (Figure 5). However, no study had reported the effect of S100 proteins on pulp tissue.

RAGE is a member of the immunoglobulin superfamily, which is mainly expressed on endothelial cells, vascular smooth muscle cells, macrophages, and monocytes [49]. RAGE is also known as a pattern recognition receptor that plays important roles in the innate immune response by NF-κB activation via binding to DAMPs including the protein S100 family [35,36]. Odontoblasts express RAGE during tooth development [50]. Moreover, RAGE expression in human pulp cells is upregulated by odontoblastic induction [51]. These reports suggest that protein S100/RAGE signaling is implicated in dentinogenesis. In addition to the present results showing that the tunnel defect was not observed in S100-A7-treated samples, protein S100-A7 might play an important role as an antimicrobial peptide that acts as a protective barrier against bacterial infection in the oral environment [52,53]. Based on these reports and ours results, we focused on protein S100-A7 and speculated that S100-A7 released from dentin may facilitate tertiary dentinogenesis as the pulpal wound healing response via S100/RAGE signaling.

To confirm the speculation, the distributions of protein S100-A7 and RAGE were investigated in a damaged pulp model [54,55]. Similar to previous reports [54,55], the regenerative process of the dentin-pulp complex beneath the cavity was observed in H-E-stained images (Figure 6B). During the healing process, protein S100-A7, RAGE, and CD146 showed unique expression patterns. Protein S100-A7 was expressed at the border between predentin and the separated odontoblast layer on day 1 (Figure 6C). Degradation of the dentin beneath the cavity (white arrows in Figure 6B) suggested that protein S100-A7 was released from digested dentin. Increasing numbers of protein S100-A7- and RAGE-positive cells were observed on day 3 and following the dentinogenic event on day 7, which might have been initiated by the S100/RAGE signaling pathway. CD146-positive cells, which were originally arranged in the subodontoblastic area at even intervals, started to show increased expression beneath the cavity after pulp damage on day 1, and an increased number of CD146-positive cells remained around tertiary dentin on day 7. Notably, dual-positive cells for RAGE and CD146 were distributed in damaged pulp (Figure 6D,E, black arrows). A population of DPSCs has been reported to express both RAGE [56,57] and CD146 [58]. RAGE is involved in recruitment of DPSCs, because RAGE-positive stem cells can response to and are attracted by its ligands [57]. The present results suggested that released protein S100-A7 is the cue for recruitment of DPSCs expressing both RAGE and CD146 via the S100-A7/RAGE axis, which initiates the following tertiary dentin formation. To elucidate the function of the S100-A7 on tertiary dentin formation, further in vitro experiments examining the effects of S100-A7 on dentin related gene and protein expression in odontoblastic cells should be performed. In addition to the pulp capping experiments, effects of S100-A7 on each wound healing process including migration, proliferation, and differentiation should be investigated using in vitro assays.

In conclusion, to the best of our knowledge, this is the first report to identify bioactive molecules released from DMCs digested by MMP, which is regarded as one aspect of biological wound healing processes of pulp tissue. Protein S100-A7 identified as a dentin-digested product by proteomic analysis promoted tertiary dentinogenesis in a direct pulp capping model, and the distribution of protein S100-A7 was confirmed beneath the cavity in the damaged pulp model. During the pulpal wound healing process in the damaged pulp model, recruitment of both RAGE and CD146-positive DPSCs suggested that released S100-A7 plays an important role in recruitment of DPSCs via the S100-A7/RAGE axis and the following tertiary dentinogenesis.

## Figures and Tables

**Figure 1 cells-08-01002-f001:**
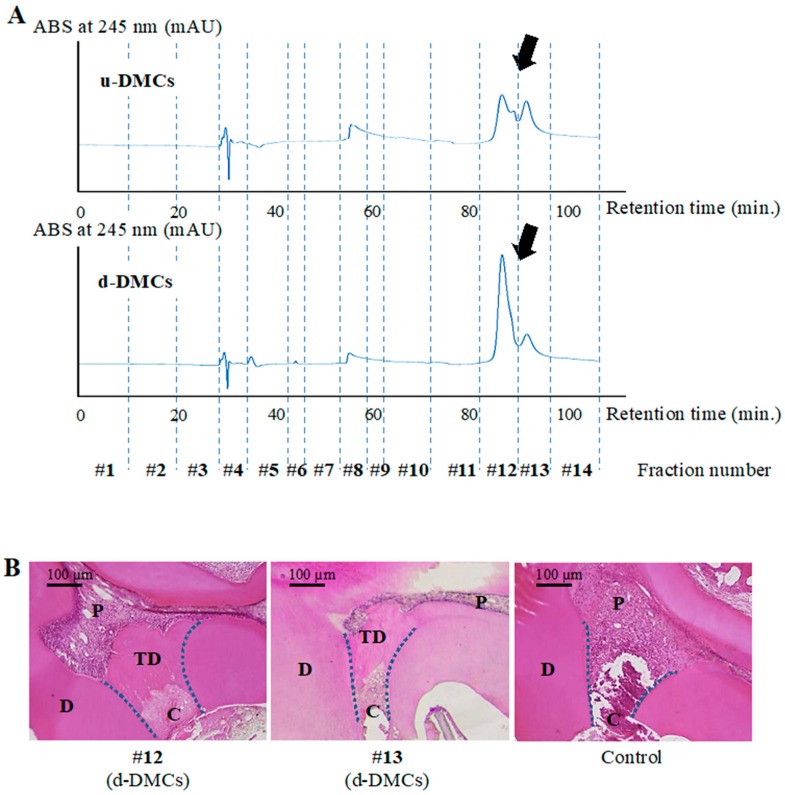
Peak analysis by reverse-phase high performance liquid chromatography (RP-HPLC) and direct pulp capping experiments for screening. (**A**) Chromatographic analysis of untreated dentin matrix components (u-DMCs) and digested DMCs (d-DMCs). Arrows indicate different shaped peaks between the samples. Each sample was divided into 14 fractions. (**B**) Histological images of tertiary dentin formation at 28 days after direct pulp capping using fractions #12 or #13 of d-DMCs or the control (D = primary dentin, TD = tertiary dentin, P = pulp, and C = cavity). Images are representative of three independent experiments.

**Figure 2 cells-08-01002-f002:**
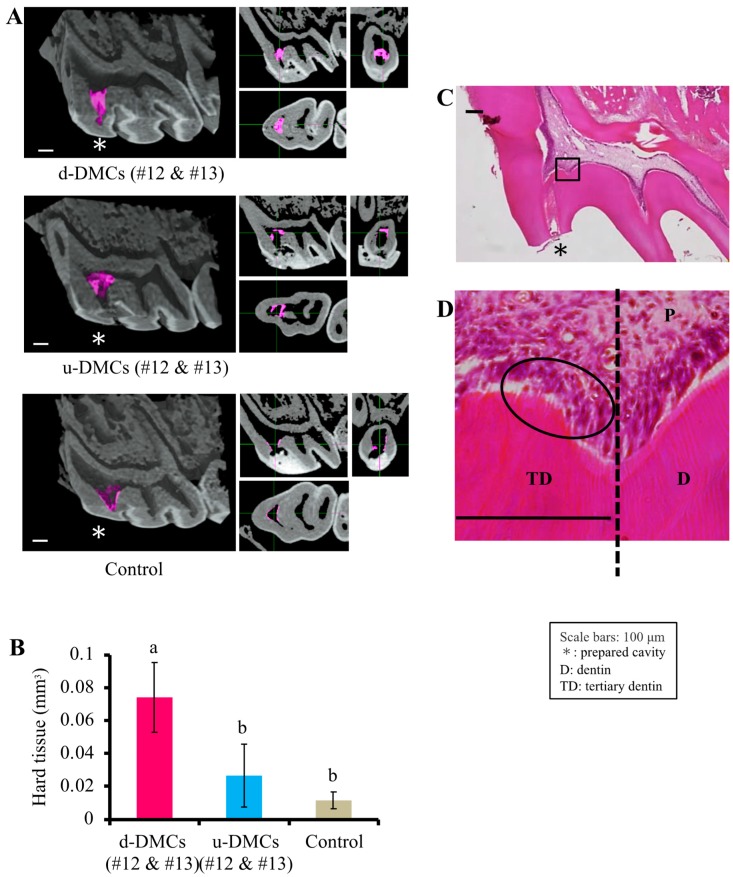
Effects of the fractions on tertiary dentinogenesis. (**A**) µ-CT images of tertiary dentin (pink area) induced using fractions #12 and #13 from d-DMCs or u-DMCs, or the control as pulp capping materials. (**B**) Quantification of the tertiary dentin volume (*n* = 3 per group). (**C**,**D**) Panoramic and high-power H-E-stained images of tertiary dentin formation promoted by d-DMC fractions. An odontoblast-like cell layer and tubular-like structure were observed in the circled area in (D) (D = primary dentin, TD = tertiary dentin, and P = pulp). Images are representative of three independent experiments. Quantitative data are means ± s.e.m. Groups with similar lowercase letters (i.e., a and b) are not significantly different (*p* > 0.05).

**Figure 3 cells-08-01002-f003:**
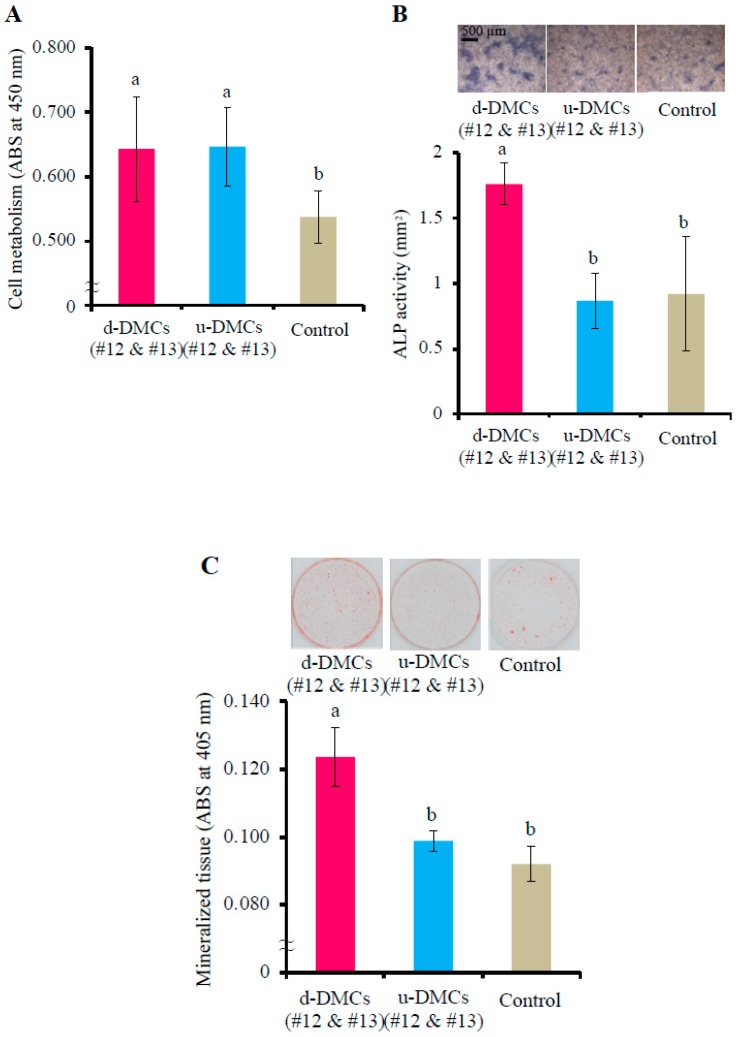
Effects of the fractions on RPPCs in vitro. Effects of fractions #12 and #13 from d-DMCs on pulp cell metabolism (**A**), alkaline phosphatase (ALP) activity (**B**), and mineralization (**C**) (*n* = 6 per group). Data are representative of five independent experiments. Quantitative data are the means ± s.e.m. Groups with similar lowercase letters (i.e., a and b) are not significantly different (*p* > 0.05).

**Figure 4 cells-08-01002-f004:**
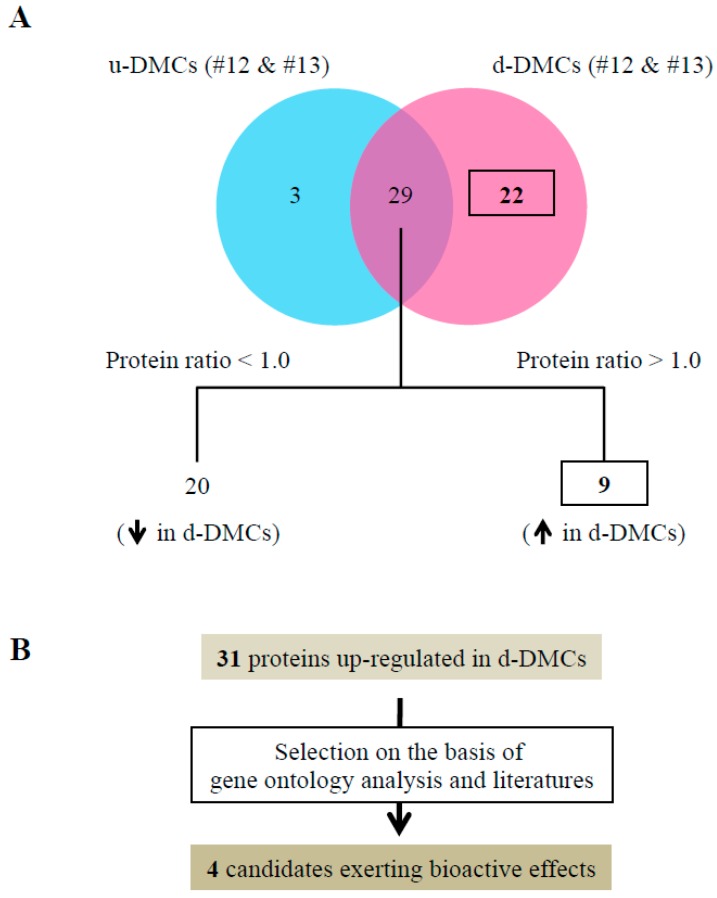
Proteomic analysis of fractions and selection of critical molecules. (**A**) Venn diagrams showing the results of protein identification of fractions #12 and #13 from u-DMCs or d-DMCs. Label-free proteome quantification was performed using MaxQuant software to calculate the protein ratio among 29 proteins common to both samples. Nine proteins showing a high protein ratio (>1) among 29 common proteins and 22 proteins unique to the fractions of d-DMCs were specifically increased in d-DMCs (protein ratios were calculated by the following formula: [protein ratio] = [intensity of protein in the fraction of d-DMCs]/[intensity of protein in the fraction of u-DMCs]). (**B**) Workflow to select candidate critical molecules for wound healing promotion. Four proteins (Table 1) were selected among 31 increased proteins based on gene ontology analysis and previous studies.

**Figure 5 cells-08-01002-f005:**
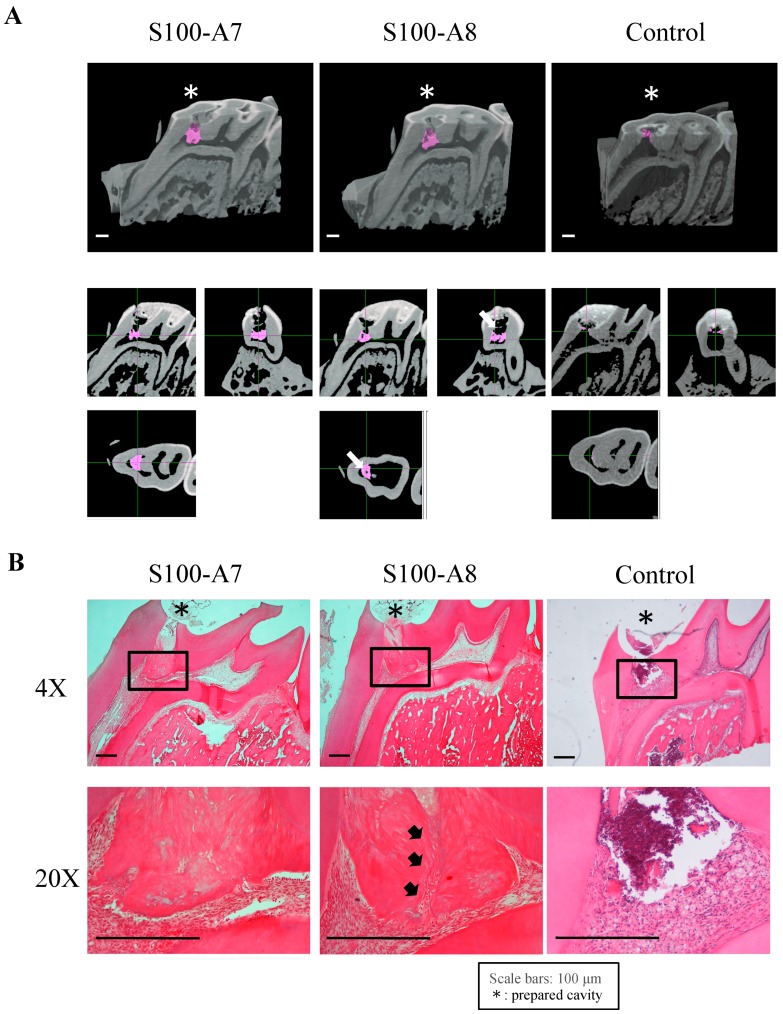
Effect of proteins S100-A7 and S100-A8 on tertiary dentinogenesis. Analyses of tertiary dentin using protein S100-A7 or S100-A8 as pulp capping materials. (**A**) Results of µ-CT. A tunnel defect in tertiary dentin is indicated by white arrows. (**B**) H-E-stained images. A tunnel defect in tertiary dentin is indicated by black arrows. Scale bar: 100 μm (* = cavity). Images are representative of three independent experiments.

**Figure 6 cells-08-01002-f006:**
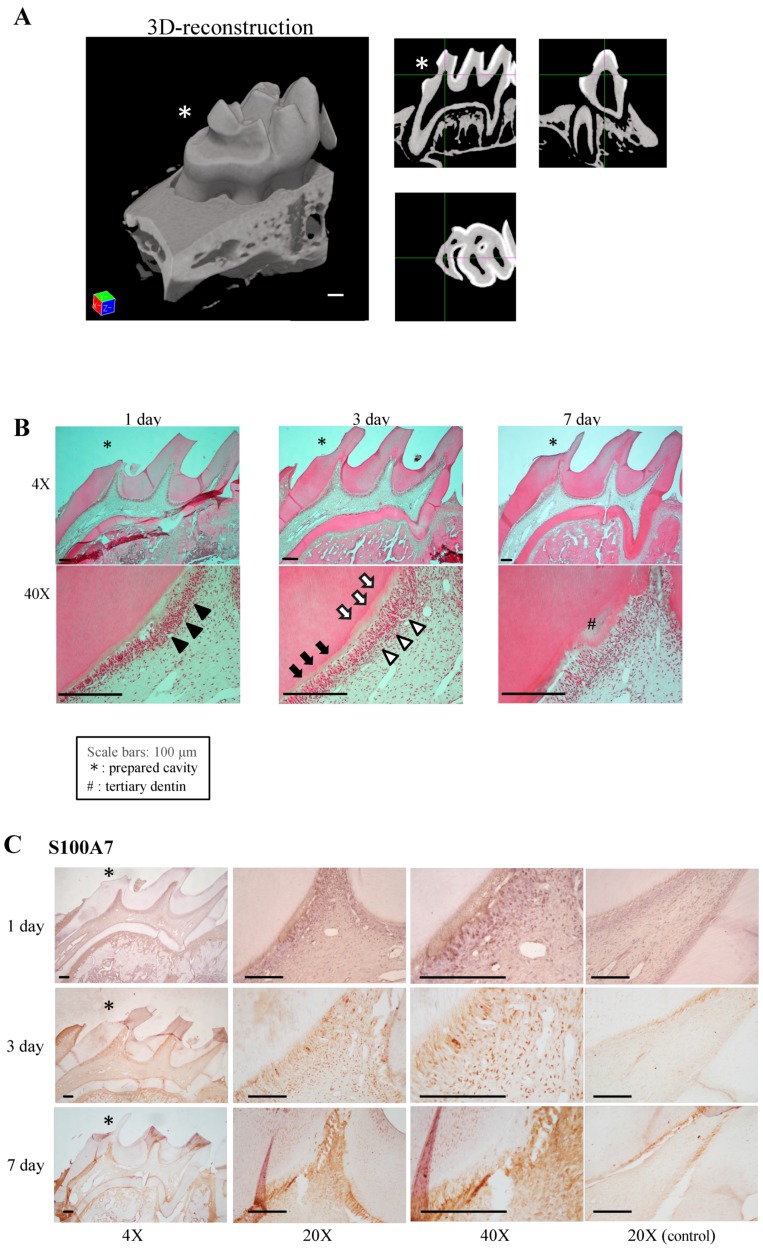
Localization of protein S100-A7-related proteins in the wound healing process. (**A**) Three-dimensional reconstruction of a cavity-prepared tooth. (**B**) Panoramic and high-power H-E-stained images of pulp tissue after cavity preparation. Detachment or disappearance of odontoblasts is indicated by black arrow heads and reattachment of newly differentiated odontoblast-like cells is indicated by white arrow heads. Black arrows indicate the ‘normal’ predentin area, and white arrows indicate the cavity affected predentin-like area. Immunohistochemical staining of S100-A7 (**C**), receptor for advanced glycation end-products (RAGE) (**D**), and CD146 (**E**). Pulp horn tissue of the non-cavity side was used as an internal control. Dual-positive cells for RAGE and CD146 are indicated by black arrows (**D**,**E**). Scale bar: 100 μm (* = prepared cavity; # = tertiary dentin). Images are representative of three independent experiments.

**Table 1 cells-08-01002-t001:** Candidates of bioactive molecules.

Protein ID	Protein Description	Protein Ration	Peptide PEP *
tr|A0A024R1X 8|A0A024R1X8_HUMAN	Junction plakoglobin	9999	0.005
sp|P31151|S10A7_HUMAN	Protein S100-A7	9999	1.095 × 10^−6^
sp|P05109|S10A8_HUMAN	Protein S100-A8	in d-DMCs only	1.270 × 10^−27^
sp|P12273|PIP_HUMAN	Prolactin-inducible protein	in d-DMCs only	0.036

Peptide PEP *: Posterior Error Probability of the identification.

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
