# Peer review of "Protein S100-A7 Derived from Digested Dentin Is a Critical Molecule for Dentin Pulp Regeneration"

_cells, 2019, doi:10.3390/cells8091002_

Round 1
Reviewer 1 Report
Based on a previous report (Okamoto et al, Scientific reports, 2017) the authors have identified the candidate molecules from digested dentin by MMP and evaluated them in Vivo for dentin regeneration. This is very interesting paper that contains novel findings in dentin repair and shows good experimental flow for the readers. However, there are some concerns which should be addressed before the publication.
1. Please discriminate the use of terminology between wound healing and tertiary dentin formation during the description of your manuscript. For example, is S100A7 important for the pulp healing or dentin formation?
2. Page10: The authors mentioned that they selected S100-A7 and S100-A8 as candidate molecules because they are regarded as components of dentin [38], [39] and bioactive proteins. In this regard, what are the other molecules, junction pla.. and prolactin.... Where did they come from? Are they not the dentin matrix component?
3. In Fig 6c, it was described that "Protein S100-A7 was secreted at the border between predentin and separated odontoblasts on day 1 after cavity preparation. On day 3, protein S100-A7-positive cells were widely detected in the pulp tissue. And, a newly differentiated odontoblast-like cell layer had started realignment on day 3?. The width of the predentin-like low calcified dentin area was increased ? On day 7, newly formed tertiary dentin was observed beneath the cavity ?" ---I can not agree with your above interpretations.
4. To elucidate the S100 function for dentin regeneration, additional in vitroexperimentsincluding "effects of S100 on DSPP, DSP, and other dentin related gene expression in odontoblastic cells" will be needed.
Reviewer 2 Report
Protein S100-A7 as a critical molecule for dentin pulp regeneration derived from digested dentin
Interesting manuscript proposing new approaches for vital pulp therapy and dental pulp regeneration. Unfortunately modest results were observed, however identifying important players for dental pulp regeneration is a plus.
Title Suggestion: Protein S100-A7 derived from digested dentin as a critical molecule for dentin pulp regeneration
Methods:
Methodology is sound. A few questions...
2. 2. Reverse phase-high performance liquid chromatography (RP-HPLC) analysis and fraction collection
“Fractions showing a certain peak or peaks were collected manually, and fractions with no peaks 89 were collected every 10 min”. Please provide more info
2. 3. 1. Pulp capping experiment
“The upper first molar was disinfected by the rubber dam 101 isolation technique and alcohol” This sentence does not look correct
“The 0.05% formic acid aqueous solution, which was used as the mobile phase buffer 108 in HPLC analysis, was diluted 50-fold with PBS and used as the control” – I believe that a more suitable control should be the use of the gelatin sponge alone or MTA/bioceramic cement which is actually considered the best materials for direct pulp capping.
2. 6. Direct pulp capping using identified proteins – Were statistical power analyses used to calculate the number of animals (N=30)?
Histological images require high magnification and better resolution
Round 2
Reviewer 1 Report
The authors addressed well the raising issues by the reviewers.